# La-MAML: Look-ahead Meta Learning for Continual Learning

**Gunshi Gupta** *
Mila, UdeM
guptagun@mila.quebec

**Karmesh Yadav** *
Carnegie Mellon University
karmeshy@andrew.cmu.edu

**Liam Paull**
Mila, UdeM
paulll@iro.umontreal.ca

## Abstract

The continual learning problem involves training models with limited capacity to perform well on a set of an unknown number of sequentially arriving tasks. While meta-learning shows great potential for reducing interference between old and new tasks, the current training procedures tend to be either slow or offline, and sensitive to many hyper-parameters. In this work, we propose *Look-ahead MAML (La-MAML)*, a fast optimisation-based meta-learning algorithm for *online*-continual learning, aided by a small episodic memory. Our proposed modulation of per-parameter learning rates in our meta-learning update allows us to draw connections to prior work on *hypergradients* and *meta-descent*. This provides a more flexible and efficient way to mitigate *catastrophic forgetting* compared to conventional *prior-based* methods. *La-MAML* achieves performance superior to other replay-based, prior-based and meta-learning based approaches for continual learning on real-world visual classification benchmarks.

## 1 Introduction

Embodied or interactive agents that accumulate knowledge and skills over time must possess the ability to continually learn. *Catastrophic forgetting* [11, 18], one of the biggest challenges in this setup, can occur when the *i.i.d.* sampling conditions required by stochastic gradient descent (SGD) are violated as the data belonging to different tasks to be learnt arrives sequentially. Algorithms for *continual learning* (CL) must also use their limited model capacity efficiently since the number of future tasks is unknown. Ensuring gradient-alignment across tasks is therefore essential, to make shared progress on their objectives. *Gradient Episodic Memory* (GEM) [17] investigated the connection between weight sharing and forgetting in CL and developed an algorithm that explicitly tried to minimise *gradient interference*. This is an objective that meta-learning algorithms implicitly optimise for (refer to [20] for derivations of the effective parameter update made in first and second order meta learning algorithms). *Meta Experience Replay* (MER) [22] formalized the transfer-interference trade-off and showed that the gradient alignment objective of GEM coincide with the objective optimised by the first order meta-learning algorithm Reptile [20].

Besides aligning gradients, meta-learning algorithms show promise for CL since they can *directly* use the meta-objective to influence model optimisation and improve on auxiliary objectives like generalisation or transfer. This avoids having to define heuristic incentives like sparsity [15] for better CL. The downside is that they are usually slow and hard to tune, effectively rendering them more suitable for *offline* continual learning [12, 22]. In this work, we overcome these difficulties and develop a gradient-based meta-learning algorithm for *efficient, online* continual learning. We first propose a base algorithm for continual meta-learning referred to as Continual-MAML (C-MAML) that utilizes a replay-buffer and optimizes a meta-objective that mitigates forgetting. Subsequently,

---

we propose a modification to C-MAML, named La-MAML, which incorporates modulation of per-parameter learning rates (LRs) to pace the learning of a model across tasks and time. Finally, we show that the algorithm is scalable, robust and achieves favourable performance on several benchmarks of varying complexity.

## 2   Related work

Relevant CL approaches can be roughly categorized into *replay-based, regularisation (or prior-based)* and *meta-learning-based* approaches.

In order to circumvent the issue of catastrophic forgetting, *replay-based methods* maintain a collection of samples from previous tasks in memory. Approaches utilising an *episodic-buffer* [5, 21] uniformly sample old data points to mimic the *i.i.d.* setup within continual learning. *Generative-replay* [27] trains generative models to be able to replay past samples, with scalability concerns arising from the difficulty of modeling complex non-stationary distributions. GEM [17] and A-GEM [6] take memory samples into account to determine altered *low-interference* gradients for updating parameters.

*Regularisation-based* methods avoid using replay at all by constraining the network weights according to heuristics intended to ensure that performance on previous tasks is preserved. This involves penalising changes to weights deemed important for old tasks [14] or enforcing weight or representational sparsity [3] to ensure that only a subset of neurons remain active at any point of time. The latter method has been shown to reduce the possibility of catastrophic interference across tasks [15, 26].

*Meta-Learning-based* approaches are fairly recent and have shown impressive results on small benchmarks like Omniglot and MNIST. MER [22], inspired by GEM[17], utilises replay to incentivise alignment of gradients between old and new tasks. Online-aware Meta Learning (OML) [12] introduces a meta-objective for a pre-training algorithm to learn an optimal representation *offline*, which is subsequently frozen and used for CL. [2, 10, 19] investigate orthogonal setups in which a learning agent uses all previously seen data to adapt quickly to an incoming stream of data, thereby ignoring the problem of catastrophic forgetting. Our motivation lies in developing a *scalable*, *online* algorithm capable of learning from limited cycles through streaming data with reduced interference on old samples. In the following sections, we review background concepts and outline our proposed algorithm. We also note interesting connections to prior work not directly pertaining to CL.

## 3   Preliminaries

We consider a setting where a sequence of $T$ tasks $[\tau_1, \tau_2, ..\tau_T]$ is learnt by observing their training data $[D_1, D_2, ..D_T]$ sequentially. We define $X^i, Y^i = \{(x_n^i, y_n^i)\}_{n=0}^{N_i}$ as the set of $N_i$ input-label pairs randomly drawn from $D_i$. An any time-step $j$ during online learning, we aim to minimize the empirical risk of the model on all the $t$ tasks seen so far ($\tau_{1:t}$), given limited access to data $(X^i, Y^i)$ from previous tasks $\tau_i$ ($i < t$). We refer to this objective as the *cumulative risk*, given by:

$$\sum_{i=1}^{t} \mathbb{E}_{(X^i, Y^i)} \left[ \ell_i \left( f_i \left( X^i; \theta \right), Y^i \right) \right] = \mathbb{E}_{(X^{1:t}, Y^{1:t})} \left[ L_t \left( f \left( X^{1:t}; \theta \right), Y^{1:t} \right) \right] \qquad (1)$$

where $\ell_i$ is the loss on $\tau_i$ and $f_i$ is a learnt, possibly task-specific mapping from inputs to outputs using parameters $\theta_0^j$. $L_t = \sum_{i=1}^{t} \ell_i$ is the sum of all task-wise losses for tasks $\tau_{1:t}$ where $t$ goes from 1 to $T$. Let $\ell$ denote some loss objective to be minimised. Then the SGD operator acting on parameters $\theta_0^j$, denoted by $U(\theta_0^j)$ is defined as:

$$U \left( \theta_0^j \right) = \theta_1^j = \theta_0^j - \alpha \nabla_{\theta_0^j} \ell(\theta_0^j) = \theta_0^j - \alpha g_0^j \qquad (2)$$

where $g_0^j = \nabla_{\theta_0^j} \ell(\theta_0^j)$. $U$ can be composed for $k$ updates as $U_k \left( \theta_0^j \right) = U... \circ U \circ U(\theta_0^j) = \theta_k^j$. $\alpha$ is a scalar or a vector LR. $U(\cdot, x)$ implies gradient updates are made on data sample $x$. We now introduce the MAML [9] and OML [12] algorithms, that we build upon in Section 4.

**Model-Agnostic Meta-Learning (MAML)**: Meta-learning [24], or *learning-to-learn* [29] has emerged as a popular approach for training models amenable to fast adaptation on limited data.

MAML [9] proposed optimising model parameters to learn a set of tasks *while* improving on auxiliary objectives like few-shot generalisation within the task distributions. We review some common terminology used in gradient-based meta-learning: 1) at a given time-step $j$ during training, model parameters $\theta_0^j$ (or $\theta_0$ for simplicity), are often referred to as an *initialisation*, since the aim is to find an ideal starting point for few-shot gradient-based adaptation on unseen data. 2) *Fast* or *inner-updates*, refer to gradient-based updates made to a copy of $\theta_0$, optimising some inner objective (in this case, $\ell_i$ for some $\tau_i$). 3) A *meta-update* involves the *trajectory* of fast updates from $\theta_0$ to $\theta_k$, followed by making a permanent gradient update (or *slow-update*) to $\theta_0$. This *slow-update* is computed by evaluating an auxiliary objective (or *meta-loss $L_{meta}$*) on $\theta_k$, and differentiating through the *trajectory* to obtain $\nabla_{\theta_0} L_{meta}(\theta_k)$. MAML thus optimises $\theta_0^j$ at time $j$, to perform optimally on tasks in $\{\tau_{1:t}\}$ after undergoing a few gradient updates on their samples. It optimises in every *meta-update*, the objective:

$$\min_{\theta_0^j} \mathbb{E}_{\tau_{1:t}} \left[ L_{meta} \left( U_k(\theta_0^j) \right) \right] = \min_{\theta_0^j} \mathbb{E}_{\tau_{1:t}} \left[ L_{meta}(\theta_k^j) \right]. \tag{3}$$

**Equivalence of Meta-Learning and CL Objectives**: The approximate equivalence of first and second-order meta-learning algorithms like Reptile and MAML was shown in [20]. MER [22] then showed that their CL objective of minimising loss on and aligning gradients between a set of tasks $\tau_{1:t}$ seen till any time $j$ *(on the left)*, can be optimised by the Reptile objective *(on the right)*, ie. :

$$\min_{\theta_0^j} \left( \sum_{i=1}^{t} \left( \ell_i(\theta_0^j) \right) - \alpha \sum_{p,q \leq t} \left( \frac{\partial \ell_p\left(\theta_0^j\right)}{\partial \theta_0^j} \cdot \frac{\partial \ell_q\left(\theta_0^j\right)}{\partial \theta_0^j} \right) \right) = \min_{\theta_0^j} \mathbb{E}_{\tau_{1:t}} \left[ L_t \left( U_k(\theta_0^j) \right) \right] \tag{4}$$

where the *meta-loss* $L_t = \sum_{i=1}^{t} \ell_i$ is evaluated on samples from tasks $\tau_{1:t}$. This implies that the procedure to meta-learn an *initialisation* coincides with learning optimal parameters for CL.

**Online-aware Meta-Learning (OML)**: [12] proposed to meta-learn a *Representation-Learning Network (RLN)* to provide a representation suitable for CL to a *Task-Learning Network (TLN)*. The RLN's representation is learnt in an *offline* phase, where it is trained using *catastrophic forgetting as the learning signal*. Data from a fixed set of tasks ($\tau_{val}$), is repeatedly used to evaluate the RLN and TLN as the TLN undergoes temporally correlated updates. In every *meta-update*'s inner loop, the TLN undergoes *fast updates* on streaming task data with a frozen RLN. The RLN and updated TLN are then evaluated through a *meta-loss* computed on data from $\tau_{val}$ along with the current task. This tests how the performance of the model has changed on $\tau_{val}$ in the process of trying to learn the streaming task. The meta-loss is then differentiated to get gradients for *slow updates* to the TLN and RLN. This composition of two losses to simulate CL in the inner loop and test *forgetting* in the outer loop, is referred to as the *OML objective*. The RLN learns to eventually provide a better representation to the TLN for CL, one which is shown to have emergent sparsity.

## 4 Proposed approach

In the previous section, we saw that the OML objective can directly regulate CL behaviour, and that MER exploits the approximate equivalence of meta-learning and CL objectives. We noted that OML trains a static representation *offline* and that MER's algorithm is prohibitively slow. We show that optimising the OML objective *online* through a multi-step MAML procedure is equivalent to a more sample-efficient CL objective. In this section, we describe *Continual-MAML* (C-MAML), the base algorithm that we propose for online continual learning. We then detail an extension to C-MAML, referred to as Look-Ahead MAML (La-MAML), outlined in Algorithm 1.

### 4.1 C-MAML

C-MAML aims to optimise the OML objective *online*, so that learning on the current task doesn't lead to forgetting on previously seen tasks. We define this objective, adapted to optimise a model's parameters $\theta$ instead of a representation at time-step $j$, as:

$$\min_{\theta_0^j} \text{OML}(\theta_0^j, t) = \min_{\theta_0^j} \sum_{\mathcal{S}_k^j \sim D_t} \left[ L_t \left( U_k(\theta_0^j, \mathcal{S}_k^j) \right) \right] \tag{5}$$

where $S_k^j$ is a stream of $k$ data tuples $\left(X_{j+l}^t, Y_{j+l}^t\right)_{l=1}^k$ from the current task $\tau_t$ that is seen by the model at time $j$. The meta-loss $L_t = \sum_{i=1}^t \ell_i$ is evaluated on $\theta_k^j = U_k(\theta_0^j, S_k^j)$. It evaluates the fitness of $\theta_k^j$ for the continual learning prediction task defined in Eq. 1 until $\tau_t$. We omit the implied data argument $(x^i, y^i) \sim (X^i, Y^i)$ that is the input to each loss $\ell_i$ in $L_t$ for any task $\tau_i$. We will show in Appendix B that optimising our objective in Eq. 5 through the $k$-step MAML update in C-MAML also coincides with optimising the CL objective of AGEM [6]:

$$
\min_{\theta_0^j} \mathbb{E}_{\tau_{1:t}} \left[ L_t \left( U_k(\theta_0^j) \right) \right] = \min_{\theta_0^j} \sum_{i=1}^t \left( \ell_i(\theta_0^j) - \alpha \frac{\partial \ell_i\left(\theta_0^j\right)}{\partial \theta_0^j} \cdot \frac{\partial \ell_t\left(\theta_0^j\right)}{\partial \theta_0^j} \right). \tag{6}
$$

This differs from Eq. 4's objective by being *asymmetric*: it focuses on aligning the gradients of $\tau_t$ and the average gradient of $\tau_{1:t}$ instead of aligning all the pair-wise gradients between tasks $\tau_{1:t}$. In Appendix D, we show empirically that gradient alignment amongst old tasks doesn't degrade while a new task is learnt, avoiding the need to repeatedly optimise the inter-task alignment between them. This results in a drastic speedup over MER's objective (Eq. 4) which tries to align all $\tau_{1:t}$ equally, thus resampling incoming samples $s \sim \tau_t$ to form a uniformly distributed batch over $\tau_{1:t}$. Since each $s$ then has $\frac{1}{t}$-th the contribution in gradient updates, it becomes necessary for MER to take multiple passes over many such uniform batches including $s$.

During training, a replay-buffer $R$ is populated through *reservoir sampling* on the incoming data stream as in [22]. At the start of every meta-update, a batch $b$ is sampled from the current task. $b$ is also combined with a batch sampled from $R$ to form the *meta-batch*, $b_m$, representing samples from both old and new tasks. $\theta_0^j$ is updated through $k$ SGD-based *inner-updates* by seeing the current task's samples from $b$ one at a time. The outer-loss or *meta-loss* $L_t(\theta_k^j)$ is evaluated on $b_m$. It indicates the performance of parameters $\theta_k^j$ on all the tasks $\tau_{1:t}$ seen till time $j$. The complete training procedure is described in Appendix C.

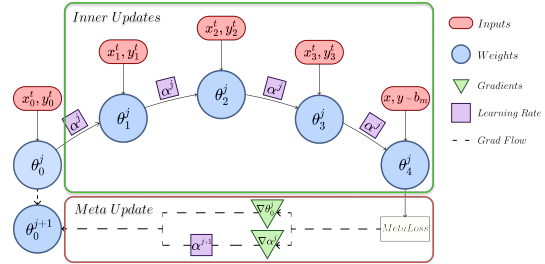

**Figure 1:** The proposed **La-MAML** algorithm: For every batch of data, the initial weights undergo a series of $k$ *fast updates* to obtain $\theta_k^j$ (here $j = 0$), which is evaluated against a meta-loss to backpropagate gradients with respect to the weights $\theta_0^0$ and LRs $\alpha^0$. First $\alpha^0$ is updated to $\alpha^1$ which is then used to update $\theta_0^0$ to $\theta_0^1$. The blue boxes indicate *fast weights* while the green boxes indicate gradients for the *slow updates*. LRs and weights are updated in an asynchronous manner.

### 4.2 La-MAML

Despite the fact that meta-learning incentivises the alignment of *within-task* and *across-task* gradients, there can still be some interference between the gradients of old and new tasks, $\tau_{1:t-1}$ and $\tau_t$ respectively. This would lead to forgetting on $\tau_{1:t-1}$, since its data is no longer fully available to us. This is especially true at the beginning of training a new task, when its gradients aren't necessarily aligned with the old ones. A mechanism is thus needed to ensure that *meta-updates* are conservative with respect to $\tau_{1:t-1}$, so as to avoid negative transfer on them. The magnitude and direction of the *meta-update* needs to be regulated, guided by how the loss on $\tau_{1:t-1}$ would be affected by the update.

We propose **Lookahead-MAML (La-MAML)**, where we include a set of learnable per-parameter learning rates (LRs) to be used in the *inner updates*, as depicted in Figure 1. This is motivated by our observation that the expression for the gradient of Eq. 5 with respect to the inner loop's LRs directly reflects the alignment between the old and new tasks. The augmented learning objective is defined as

$$
\min_{\theta_0^j, \alpha^j} \sum_{S_k^j \sim D_t} \left[ L_t \left( U_k \left( \alpha^j, \theta_0^j, S_k^j \right) \right) \right], \tag{7}
$$

and the gradient of this objective at time $j$, with respect to the LR vector $\alpha^j$ (denoted as $g_{MAML}(\alpha^j)$) is then given as:

$$g_{MAML}(\alpha^j) = \frac{\partial}{\partial \alpha^j} L_t\left(\theta_k^j\right) = \frac{\partial}{\partial \theta_k^j} L_t\left(\theta_k^j\right) \cdot \left(-\sum_{k'=0}^{k-1} \frac{\partial}{\partial \theta_{k'}^j} \ell_t\left(\theta_{k'}^j\right)\right). \tag{8}$$

We provide the full derivation in the Appendix A, and simply state the expression for a first-order approximation [9] of $g_{MAML}(\alpha)$ here. The first term in $g_{MAML}(\alpha)$ corresponds to the gradient of the meta-loss on batch $b_m$: $g_{meta}$. The second term indicates the cumulative gradient from the inner-updates: $g_{traj}$. This expression indicates that the gradient of the LRs will be negative when the inner product between $g_{meta}$ and $g_{traj}$ is high, ie. the two are aligned; zero when the two are orthogonal (not interfering) and positive when there is interference between the two. Negative (positive) LR gradients would pull up (down) the LR magnitude. We depict this visually in Figure 2.

---

**Algorithm 1** La-MAML : Look-ahead MAML

**Input:** Network weights $\theta$, LRs $\alpha$, inner objective $\ell$, meta objective $L$, learning rate for $\alpha$ : $\eta$
$j \leftarrow 0, R \leftarrow \{\}$ ▷ Initialise Replay Buffer
**for** $t := 1$ **to** $T$ **do**
    **for** $ep := 1$ **to** $num_{epochs}$ **do**
        **for** batch $b$ **in** $(X^t, Y^t) \sim D_t$ **do**
            $k \leftarrow sizeof(b)$
            $b_m \leftarrow Sample(R) \cup b$
            **for** $n = 0$ **to** $k-1$ **do**
                Push $b[k']$ to R with reservoir sampling
                $\theta_{k'+1}^j \leftarrow \theta_{k'}^j - \alpha^j \cdot \nabla_{\theta_{k'}^j}$
            **end for**
            $\alpha^{j+1} \leftarrow \alpha^j - \eta \nabla_{\alpha^j} L_t(\theta_k^j, b_m)$
            $\theta_0^{j+1} \leftarrow \theta_0^j - max(0, \alpha^{j+1}) \cdot \nabla_{\theta_0^j} L_t(\theta_k^j, b_m)$
            $j \leftarrow j + 1$
        **end for**
    **end for**
**end for**

(a)
(b)

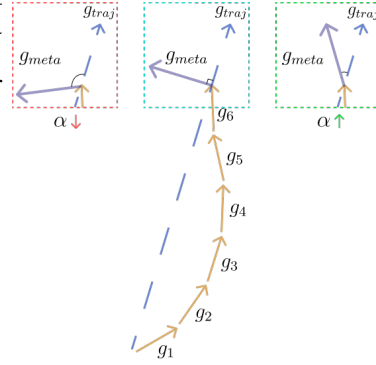

**Figure 2:** Different scenarios for the alignment of $g_{traj}$ (blue dashed line) and $g_{meta}$, going from interference (left) to alignment (right). Yellow arrows denote the *inner updates*. The LR $\alpha$ increases (decreases) when gradients align (interfere).

We propose updating the network weights and LRs *asynchronously* in the meta-update. Let $\alpha^{j+1}$ be the updated LR vector obtained by taking an SGD step with the LR gradient from Eq. 8 at time $j$. We then update the weights as:

$$\theta_0^{j+1} \leftarrow \theta_0^j - max(0, \alpha^{j+1}) \cdot \nabla_{\theta_0^j} L_t(\theta_k^j) \tag{9}$$

where $k$ is the number of steps taken in the inner-loop. The LRs $\alpha^{j+1}$ are clipped to positive values to avoid *ascending* the gradient, and also to avoid making *interfering* parameter-updates, thus mitigating catastrophic forgetting. The meta-objective thus conservatively modulates the pace and direction of learning to achieve quicker learning progress on a new task while facilitating transfer on old tasks. Algorithm 1 [2] illustrates this procedure. Lines (a), (b) are the only difference between C-MAML and La-MAML, with C-MAML using a fixed scalar LR $\alpha$ for the meta-update to $\theta_0^j$ instead of $\alpha^{j+1}$.

Our meta-learning based algorithm incorporates concepts from both prior-based and replay-based approaches. The LRs modulate the parameter updates in an data-driven manner, guided by the interplay between gradients on the replay samples and the streaming task. However, since LRs evolve with every meta-update, their decay is temporary. This is unlike many prior-based approaches, where penalties on the change in parameters gradually become so high that the network capacity saturates [14]. Learnable LRs can be modulated to high and low values as tasks arrive, thus being a simpler, flexible and elegant way to constrain weights. This asynchronous update resembles trust-region optimisation [31] or *look-ahead search* since the step-sizes for each parameter are adjusted based on

the loss incurred after applying hypothetical updates to them. Our LR update is also analogous to the heuristic uncertainty-based LR update in UCB [8], BGD [32], which we compare to in Section 5.3.

## 4.3 Connections to Other Work

**Stochastic Meta-Descent (SMD)**: When learning over a non-stationary data distribution, using decaying LR schedules is not common. Strictly diminishing LR schedules aim for closer *convergence* to a fixed mimima of a stationary distribution, which is at odds with the goal of online learning. It is also not possible to manually tune these schedules since the extent of the data distribution is unknown. However, *adaptivity* in LRs is still highly desired to adapt to the optimisation landscape, accelerate learning and modulate the degree of adaptation to reduce catastrophic forgetting. Our adaptive LRs can be connected to work on *meta-descent* [4, 25] in *offline* supervised learning (OSL). While several variations of *meta-descent* exist, the core idea behind them and our approach is *gain adaptation*. While we adapt the gain based on the correlation between old and new task gradients to make shared progress on all tasks, [4, 25] use the correlation between two successive stochastic gradients to converge faster. We rely on the meta-objective's differentiability with respect to the LRs, to obtain LR *hypergradients* automatically.

**Learning LRs in meta-learning**: Meta-SGD [16] proposed learning the LRs in MAML for few-shot learning. Some notable differences between their update and ours exist. They *synchronously* update the weights and LRs while our *asynchronous* update to the LRs serves to carry out a more conservative update to the weights. The intuition for our update stems from the need to mitigate gradient interference and its connection to the transfer-interference trade-off ubiquitous in continual learning. $\alpha$-MAML [28] analytically updates the two *scalar* LRs in the MAML update for more adaptive few-shot learning. Our *per-parameter* LRs are modulated implicitly through back-propagation, to regulate change in parameters based on their alignment across tasks, providing our model with a more powerful degree of adaptability in the CL domain.

## 5 Experiments

In this section, we evaluate La-MAML in settings where the model has to learn a set of sequentially streaming classification tasks. *Task-agnostic* experiments, where the task identity is unknown at training and test-time, are performed on the MNIST benchmarks with a *single-headed* model. *Task-aware* experiments with known task identity, are performed on the CIFAR and TinyImagenet [1] datasets with a *multi-headed* model. Similar to [22], we use the retained accuracy (RA) metric to compare various approaches. RA is the average accuracy of the model across tasks at the end of training. We also report the *backward-transfer and interference* (BTI) values which measure the average change in the accuracy of each task from when it was learnt to the end of the last task. A smaller BTI implies lesser forgetting during training.

*Efficient Lifelong Learning (LLL)*: Formalized in [6], the setup of efficient lifelong learning assumes that incoming data for every task has to be processed in only one single pass: once processed, data samples are not accessible anymore unless they were added to a replay memory. We evaluate our algorithm on this challenging *(Single-Pass)* setup as well as the standard *(Multiple-Pass)* setup, where

**Table 1:** RA, BTI and their standard deviation on MNIST benchmarks. Each experiment is run with 5 seeds.

| METHOD | ROTATIONS | | PERMUTATIONS | | MANY | |
|---|---|---|---|---|---|---|
| | RA | BTI | RA | BTI | RA | BTI |
| ONLINE | $53.38 \pm 1.53$ | $-5.44 \pm 1.70$ | $55.42 \pm 0.65$ | $-13.76 \pm 1.19$ | $32.62 \pm 0.43$ | $-19.06 \pm 0.86$ |
| EWC | $57.96 \pm 1.33$ | $-20.42 \pm 1.60$ | $62.32 \pm 1.34$ | $-13.32 \pm 2.24$ | $33.46 \pm 0.46$ | $-17.84 \pm 1.15$ |
| GEM | $67.38 \pm 1.75$ | $-18.02 \pm 1.99$ | $55.42 \pm 1.10$ | $-24.42 \pm 1.10$ | $32.14 \pm 0.50$ | $-23.52 \pm 0.87$ |
| MER | $\mathbf{77.42} \pm \mathbf{0.78}$ | $\mathbf{-5.60} \pm \mathbf{0.70}$ | $73.46 \pm 0.45$ | $-9.96 \pm 0.45$ | $47.40 \pm 0.35$ | $-17.78 \pm 0.39$ |
| C-MAML | $77.33 \pm 0.29$ | $-7.88 \pm 0.05$ | $\mathbf{74.54} \pm \mathbf{0.54}$ | $-10.36 \pm 0.14$ | $47.29 \pm 1.21$ | $-20.86 \pm 0.95$ |
| SYNC | $74.07 \pm 0.58$ | $-6.66 \pm 0.44$ | $70.54 \pm 1.54$ | $-14.02 \pm 2.14$ | $44.48 \pm 0.76$ | $-24.18 \pm 0.65$ |
| LA-MAML | $\mathbf{77.42} \pm \mathbf{0.65}$ | $-8.64 \pm 0.403$ | $74.34 \pm 0.67$ | $\mathbf{-7.60} \pm \mathbf{0.51}$ | $\mathbf{48.46} \pm \mathbf{0.45}$ | $\mathbf{-12.96} \pm \mathbf{0.073}$ |

ideally offline training-until-convergence is performed for every task, once we have access to the data.

## 5.1 Continual learning benchmarks

First, we carry out experiments on the toy continual learning benchmarks proposed in prior CL works. **MNIST Rotations**, introduced in [17], comprises tasks to classify MNIST digits rotated by a different common angle in [0, 180] degrees in each task. In **MNIST Permutations**, tasks are generated by shuffling the image pixels by a fixed random permutation. Unlike Rotations, the input distribution of each task is unrelated

**Table 2:** Running times for MER and La-MAML on MNIST benchmarks for one epoch

| METHOD | ROTATIONS | PERMUTATIONS |
|---|---|---|
| LA-MAML | $45.95 \pm 0.38$ | $46.13 \pm 0.42$ |
| MER | $218.03 \pm 6.44$ | $227.11 \pm 12.12$ |

here, leading to less positive transfer between tasks. Both MNIST Permutation and MNIST Rotation have 20 tasks with 1000 samples per task. **Many Permutations**, a more complex version of Permutations, has five times more tasks (100 tasks) and five times less training data (200 images per task). Experiments are conducted in the low data regime with only 200 samples for Rotation and Permutation and 500 samples for Many, which allows the differences between the various algorithm to become prominent (detailed in Appendix G). We use the same architecture and experimental settings as in MER [22], allowing us to compare directly with their results. We use the cross-entropy loss as the *inner* and *outer* objectives during meta-training. Similar to [20], we see improved performance when evaluating and summing the *meta-loss* at all steps of the inner updates as opposed to just the last one.

We compare our method in the *Single-Pass* setup against multiple baselines including *Online*, *Independent*, *EWC* [14], *GEM* [17] and *MER* [22] (detailed in Appendix H), as well as different ablations (discussed in Section 5.3). In Table 1, we see that La-MAML achieves comparable or better performance than the baselines on all benchmarks. Table 2 shows that La-MAML matches the performance of MER in less than 20% of the training time, owing to its sample-efficient objective which allows it to make make more learning progress per iteration. This also allows us to scale it to real-world visual recognition problems as described next.

## 5.2 Real-world classification

While La-MAML fares well on the MNIST benchmarks, we are interested in understanding its capabilities on more complex visual classification benchmarks. We conduct experiments on the **CIFAR-100** dataset in a task-incremental manner [17] where, 20 tasks comprising of disjoint *5-way* classification problems are streamed. We also evaluate on the **TinyImagenet-200** dataset by partitioning its 200 classes into 40 *5-way* classification tasks. Experiments are carried out in both the *Single-Pass* and *Multiple-Pass* settings, where in the latter we allow all CL approaches to train up to a maximum of 10 epochs. Each method is allowed a replay-buffer, containing upto 200 and 400 samples for CIFAR-100 and TinyImagenet respectively. We provide further details about the baselines in Appendix H and about the architectures, evaluation setup and hyper-parameters in Appendix G.

Table 3 reports the results of these experiments. We consistently observe superior performance of La-MAML as compared to other CL baselines on both datasets across setups. While the iCARL baseline attains lower BTI in some setups, it achieves that at the cost of much lower performance throughout learning. Among the high-performing approaches, La-MAML has the lowest BTI. Recent work [7, 22] noted that Experience Replay (ER) is often a very strong baseline that closely matches the performance of the proposed algorithms. We highlight the fact that meta-learning and LR modulation combined show an improvement of more than 10 and 18% (as the number of tasks increase from CIFAR to TinyImagenet) over the ER baseline in our case, with limited replay. Overall, we see that our method is robust and better-performing under both the standard and LLL setups of CL which come with different kinds of challenges. Many CL methods [8, 26] are suitable for only one of the two setups. Further, as explained in Figure 3, our model evolves to become resistant to *forgetting* as training progresses. This means that beyond a point, it can keep making gradient updates on a small window of incoming samples without needing to do *meta-updates*.

## 5.3 Evaluation of La-MAML's learning rate modulation

To capture the gains from learning the LRs, we compare La-MAML with our base algorithm, **C-MAML**. We ablate our choice of updating LRs asynchronously by constructing a version of C-MAML where per-parameter learnable LRs are used in the inner updates while the meta-update still uses a constant scalar LR during training. We refer to it as *Sync-La-MAML* or **Sync** since it has synchronously updated LRs that don't modulate the meta-update. We also construct an ablation referred to as *La-ER*, where the parameter updates are carried out as in ER but the LRs are modulated using the La-MAML objective's first-order version. This tells us what the gains of LR modulation are over ER, since there is no meta-learning to encourage gradient alignment of the model parameters. While only minor gains are seen on the MNIST benchmarks from asynchronous LR modulation, the performance gap increases as the tasks get harder. On CIFAR-100 and TinyImagenet, we see a trend in the RA of our variants with La-MAML performing best followed by *Sync*. This shows that optimising the LRs aids learning and our *asynchronous* update helps in knowledge consolidation by enforcing conservative updates to mitigate interference.

To test our LR modulation against an alternative *bayesian* modulation scheme proposed in BGD [32], we define a baseline called Meta-BGD where per-parameter variances are modulated instead of LRs. This is described in further detail in Appendix H. Meta-BGD emerges as a strong baseline and matches the performance of C-MAML given enough Monte Carlo iterations $m$, implying $m$ times more computation than C-MAML. Additionally, Meta-BGD was found to be sensitive to hyperparameters and required extensive tuning. We present a discussion of the robustness of our approach in Appendix E, as well as a discussion of the setups adopted in prior work, in Appendix I.

We also compare the gradient alignment of our three variants along with ER in Table 4 by calculating the cosine similarity between the gradients of the replay samples and newly arriving data samples. As previously stated, the aim of many CL algorithms is to achieve high gradient alignment across tasks to allow parameter-sharing between them. We see that our variants achieve an order of magnitude higher cosine similarity compared to ER, verifying that our objective promotes gradient alignment.

## 6 Conclusion

We introduced La-MAML, an efficient meta-learning algorithm that leverages replay to avoid forgetting and favor positive backward transfer by learning the weights and LRs in an asynchronous manner. It is capable of learning online on a non-stationary stream of data and scales to vision tasks. We presented results that showed better performance against the state-of-the-art in the setup of efficient lifelong learning (LLL) [6], as well as the standard continual learning setting. In the future, more work on analysing and producing good optimizers for CL is needed, since many of our standard go-to optimizers like Adam [13] are primarily aimed at ensuring faster convergence in *stationary* supervised learning setups. Another interesting direction is to explore how the connections to *meta-descent* can lead to more stable training procedures for meta-learning that can automatically adjust hyper-parameters on-the-fly based on training dynamics.

**Table 3:** Results on the standard continual (Multiple) and LLL (Single) setups with CIFAR-100 and TinyImagenet-200. Experiments are run with 3 seeds. * indicates result omitted due to high instability.

| METHOD | CIFAR-100 | | | | TINYIMAGENET | | | |
| --- | --- | --- | --- | --- | --- | --- | --- | --- |
| | MULTIPLE | | SINGLE | | MULTIPLE | | SINGLE | |
| | RA | BTI | RA | BTI | RA | BTI | RA | BTI |
| IID | $85.60 \pm 0.40$ | - | - | - | $77.1 \pm 1.06$ | - | - | - |
| ER | $59.70 \pm 0.75$ | $-16.50 \pm 1.05$ | $47.88 \pm 0.73$ | $-12.46 \pm 0.83$ | $48.23 \pm 1.51$ | $-19.86 \pm 0.70$ | $39.38 \pm 0.38$ | $-14.33 \pm 0.89$ |
| iCARL | $60.47 \pm 1.09$ | $-15.10 \pm 1.04$ | $53.55 \pm 1.69$ | $\mathbf{-8.03 \pm 1.16}$ | $54.77 \pm 0.32$ | $\mathbf{-3.93 \pm 0.55}$ | $45.79 \pm 1.49$ | $\mathbf{-2.73 \pm 0.45}$ |
| GEM | $62.80 \pm 0.55$ | $-17.00 \pm 0.26$ | $48.27 \pm 1.10$ | $-13.7 \pm 0.70$ | $50.57 \pm 0.61$ | $-20.50 \pm 0.10$ | $40.56 \pm 0.79$ | $-13.53 \pm 0.65$ |
| AGEM | $58.37 \pm 0.13$ | $-17.03 \pm 0.72$ | $46.93 \pm 0.31$ | $-13.4 \pm 1.44$ | $46.38 \pm 1.34$ | $-19.96 \pm 0.61$ | $38.96 \pm 0.47$ | $-13.66 \pm 1.73$ |
| MER | - | - | $51.38 \pm 1.05$ | $-12.83 \pm 1.44$ | - | - | $44.87 \pm 1.43$ | $-12.53 \pm 0.58$ |
| META-BGD | $65.09 \pm 0.77$ | $-14.83 \pm 0.40$ | $57.44 \pm 0.95$ | $-10.6 \pm 0.45$ | * | * | $50.64 \pm 1.98$ | $-6.60 \pm 1.73$ |
| C-MAML | $65.44 \pm 0.99$ | $-13.96 \pm 0.86$ | $55.57 \pm 0.94$ | $-9.49 \pm 0.45$ | $61.93 \pm 1.55$ | $-11.53 \pm 1.11$ | $48.77 \pm 1.26$ | $-7.6 \pm 0.52$ |
| LA-ER | $67.17 \pm 1.14$ | $-12.63 \pm 0.60$ | $56.12 \pm 0.61$ | $-7.63 \pm 0.90$ | $54.76 \pm 1.94$ | $-15.43 \pm 1.36$ | $44.75 \pm 1.96$ | $-10.93 \pm 1.32$ |
| SYNC | $67.06 \pm 0.62$ | $-13.66 \pm 0.50$ | $58.99 \pm 1.40$ | $-8.76 \pm 0.95$ | $65.40 \pm 1.40$ | $-11.93 \pm 0.55$ | $\mathbf{52.84 \pm 2.55}$ | $-7.3 \pm 1.93$ |
| LA-MAML | $\mathbf{70.08 \pm 0.66}$ | $\mathbf{-9.36 \pm 0.47}$ | $\mathbf{61.18 \pm 1.44}$ | $-9.00 \pm 0.2$ | $\mathbf{66.99 \pm 1.65}$ | $\mathbf{-9.13 \pm 0.90}$ | $52.59 \pm 1.35$ | $\mathbf{-3.7 \pm 1.22}$ |

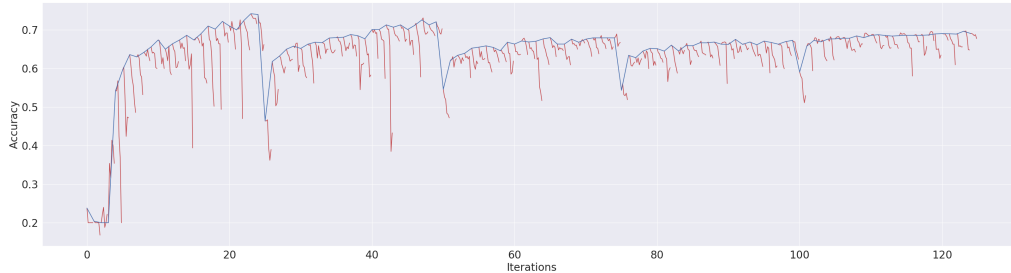

**Figure 3:** Retained Accuracy (RA) for La-MAML plotted every 25 meta-updates up to Task 5 on CIFAR-100. RA at iteration $j$ (with $j$ increasing along the x-axis) denotes accuracy on all tasks seen uptil then. Red denotes the RA computed during the *inner updates* (at $\theta_k^j$). Blue denotes RA computed at $\theta_0^{j+1}$ right after a *meta-update*. We see that in the beginning, inner updates lead to catastrophic forgetting (CF) since the weights are not suitable for CL yet, but eventually become resistant when trained to retain old knowledge while learning on a stream of correlated data. We also see that RA maintains its value even as more tasks are added indicating that the model is successful at learning new tasks without sacrificing performance on old ones.

**Table 4:** Gradient Alignment on CIFAR-100 and TinyImagenet dataset (values lie in [-1,1], higher is better)

| DATASET | ER | C-MAML | SYNC | LA-MAML |
|---|---|---|---|---|
| CIFAR-100 | $0.22 \times 10^{-2}$ $_{\pm 0.0017}$ | $1.84 \times 10^{-2}$ $_{\pm 0.0003}$ | $2.28 \times 10^{-2}$ $_{\pm 0.0004}$ | $1.86 \times 10^{-2}$ $_{\pm 0.0027}$ |
| TINYIMAGENET | $0.27 \times 10^{-2}$ $_{\pm 0.0005}$ | $1.74 \times 10^{-2}$ $_{\pm 0.0005}$ | $2.17 \times 10^{-2}$ $_{\pm 0.0020}$ | $2.14 \times 10^{-2}$ $_{\pm 0.0023}$ |

## Broader Impact

This work takes a step towards enabling deployed models to operate while learning *online*. This would be very relevant for online, interactive services like recommender systems or home robotics, among others. By tackling the problem of catastrophic forgetting, the proposed approach goes some way in allowing models to add knowledge incrementally without needing to be re-trained from scratch. Training from scratch is a compute intensive process, and even requires access to data that might not be available anymore. This might entail having to navigate a privacy-performance trade-off since many techniques like federated learning actually rely on not having to share data across servers, in order to protect user-privacy.

The proposed algorithm stores and replays random samples of prior data, and even with the higher alignment of the samples within a task under the proposed approach, there will eventually be some concept drift. While the proposed algorithm itself does not rely on or introduce any biases, any bias in the sampling strategy itself might influence the distribution of data that the algorithm *remembers* and performs well on.

## Acknowledgments and Disclosure of Funding

The authors are grateful to Matt Riemer, Sharath Chandra Raparthy, Alexander Zimin, Heethesh Vhavle and the anonymous reviewers for proof-reading the paper and suggesting improvements. This research was enabled in part by support provided by Compute Canada (`www.computecanada.ca`).

## Footnotes

[2]The code for our algorithm can be found at: `https://github.com/montrealrobotics/La-MAML`

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
