[Supplementary Material]

# A  Hypergradient Derivation for La-MAML

We derive the gradient of the weights $\theta_0^j$ and LRs $\alpha^j$ at time-step $j$ under the $k$-step MAML objective, with $L_t = \sum_{i=0}^{t} \ell_i$ as the *meta-loss* and $\ell_t$ as the *inner-objective*:

$$g_{\text{MAML}}(\alpha^j) = \frac{\partial}{\partial \alpha^j} L_t\left(\theta_k^j\right) = \frac{\partial}{\partial \theta_k^j} L_t\left(\theta_k^j\right) \cdot \frac{\partial}{\partial \alpha^j}\left(\theta_k^j\right)$$

$$= \frac{\partial}{\partial \theta_k^j} L_t\left(\theta_k^j\right) \cdot \frac{\partial}{\partial \alpha^j}\left(U\left(\theta_{k-1}^j\right)\right)$$

$$= \frac{\partial}{\partial \theta_k^j} L_t\left(\theta_k^j\right) \cdot \frac{\partial}{\partial \alpha^j}\left(\theta_{k-1}^j - \alpha^j \frac{\partial \ell_t(\theta_{k-1}^j)}{\partial \theta_{k-1}^j}\right)$$

$$= \frac{\partial}{\partial \theta_k^j} L_t\left(\theta_k^j\right) \cdot \left(\frac{\partial}{\partial \alpha^j}\theta_{k-1}^j - \frac{\partial}{\partial \alpha^j}\left(\alpha^j \frac{\partial \ell_t(\theta_{k-1}^j)}{\partial \theta_{k-1}^j}\right)\right)$$

$$= \frac{\partial}{\partial \theta_k^j} L_t\left(\theta_k^j\right) \cdot \left(\frac{\partial}{\partial \alpha^j}\theta_{k-1}^j - \frac{\partial \ell_t(\theta_{k-1}^j)}{\partial \theta_{k-1}^j}\right)$$

(Taking $\dfrac{\partial \ell_t\left(\theta_{k-1}^j\right)}{\partial \theta_{k-1}^j}$ as a constant w.r.t $\alpha^j$ to get the first-order MAML approximation)

$$= \frac{\partial}{\partial \theta_k^j} L_t\left(\theta_k^j\right) \cdot \left(\frac{\partial}{\partial \alpha^j}U\left(\theta_{k-2}^j\right) - \left(\frac{\partial \ell_t(\theta_{k-1}^j)}{\partial \theta_{k-1}^j}\right)\right)$$

$$= \frac{\partial}{\partial \theta_k^j} L_t\left(\theta_k^j\right) \cdot \left(\frac{\partial}{\partial \alpha^j}\theta_0^j - \sum_{k'=0}^{k-1} \frac{\partial \ell_t(\theta_n^j)}{\partial \theta_n^j}\right) \quad (a)$$

$$= \frac{\partial}{\partial \theta_k^j} L_t\left(\theta_k^j\right) \cdot \left(-\sum_{k'=0}^{k-1} \frac{\partial \ell_t(\theta_{k'}^j)}{\partial \theta_{k'}^j}\right) \quad (b)$$

Where (a) is obtained by recursively expanding and differentiating the update function $U()$ as done in the step before it. (b) is obtained by assuming that the initial weight in the meta-update at time j : $\theta_0^j$, is constant with respect to $\alpha^j$.

Similarly we can derive the MAML gradient for the weights $\theta_0^j$, denoted as $g_{\text{MAML}}(\theta_0^j)$ as:

$$g_{\text{MAML}}(\theta_0^j) = \frac{\partial}{\partial \theta_0^j} L_t(\theta_k^j) = \frac{\partial}{\partial \theta_k^j} L_t(\theta_k^j) \frac{\partial \theta_k^j}{\partial \theta_0^j} = \frac{\partial}{\partial \theta_k^j} L_t(\theta_k^j) \frac{\partial U_k(\theta_{k-1}^j)}{\partial \theta_0^j}$$

$$= \frac{\partial}{\partial \theta_k^j} L_t(\theta_k^j) \frac{\partial}{\partial \theta_{k-1}^j} U(\theta_{k-1}^j) \cdots \frac{\partial}{\partial \theta_0^j} U(\theta_1^j)$$

(repeatedly applying chain rule and using $\theta_k^j = U(\theta_{k-1}^j)$ )

$$= L_t'(\theta_k^j)\left(I - \alpha \ell_t''(\theta_{k-1}^j)\right) \cdots \left(I - \alpha \ell_t''(\theta_0^j)\right)$$

$$\left(\text{using } U'(\theta_{k'}^j) = I - \alpha \ell_t''(\theta_{k'}^j)\right) \quad (' \text{ implies derivative with respect to argument})$$

$$= \left(\prod_{k'=0}^{k-1}\left(I - \alpha \ell_t''(\theta_{k'}^j)\right)\right) L_t'(\theta_k^j)$$

Setting all first-order gradient terms as constants to ignore second-order derivatives, we get the first order approximation as:

$$g_{\text{FOMAML}}(\theta_0^j) = \left(\prod_{k'=0}^{k-1}\left(I - \alpha \ell_t''\left(\theta_{k'}^j\right)\right)\right) L_t'(\theta_k^j) = L_t'(\theta_k^j)$$

In Appendix B, we show the equivalence of the C-MAML and CL objectives in Eq. 6 by showing that the gradient of the former ($g_{\text{MAML}}(\theta_0^j)$) is equivalent to the gradient of the latter.

# B Equivalence of Objectives

It is straightforward to show that when we optimise the OML objective through the $k$-step MAML update, as proposed in C-MAML in Eq. 5:

$$\min_{\theta_0^j} \mathbb{E}_{\tau_{1:t}} \left[ L_t \left( U_k(\theta_0^j) \right) \right] \tag{10}$$

where the *inner-updates* are taken using data from the streaming task $\tau_t$, and the *meta-loss* $L_t(\theta) = \sum_{i=1}^t \ell_i(\theta)$ is computed on the data from all tasks seen so far, it will correspond to minimising the following surrogate loss used in CL :

$$\min_{\theta_0^j} \sum_{i=1}^t \left( \ell_i(\theta_0^j) - \alpha \frac{\partial \ell_i \left( \theta_0^j \right)}{\partial \theta_0^j} \cdot \frac{\partial \ell_t \left( \theta_0^j \right)}{\partial \theta_0^j} \right) \tag{11}$$

We show the equivalence for the case when $k = 1$, for higher $k$ the form gets more complicated but essentially has a similar set of terms. Reptile [20] showed that the $k$-step MAML gradient for the weights $\theta_0^j$ at time $j$, denoted as $g_{\text{MAML}}(\theta_0^j)$ is of the form:

$$\frac{\partial L_{meta}(\theta_k^j)}{\partial \theta_0^j} = \bar{g}_k - \alpha \bar{H}_k \sum_{k'=0}^{k-1} \bar{g}_{k'} - \alpha \sum_{k'=0}^{k-1} \bar{H}_{k'} \bar{g}_k + O\left(\alpha^2\right) \quad (\alpha \text{ is the } \textit{inner-loop} \text{ learning rate})$$

$$= \bar{g}_1 - \alpha \bar{H}_1 \bar{g}_0 - \alpha \bar{H}_0 \bar{g}_1 + O\left(\alpha^2\right) \quad (\text{using } k = 1)$$

Expressing the terms as derivatives, and using $\frac{\partial}{\partial \theta_0^j} (\bar{g}_0 \cdot \bar{g}_1) = \bar{H}_1 \bar{g}_0 + \bar{H}_0 \bar{g}_1$, we get :

$$= \frac{\partial L_{meta}(\theta_0^j)}{\partial \theta_0^j} - \frac{\partial}{\partial \theta_0^j} (\bar{g}_0 \cdot \bar{g}_1)$$

$$= \frac{\partial \left( \sum_{i=1}^t \ell_i(\theta_0^j) - \alpha \bar{g}_1 \cdot \bar{g}_0 \right)}{\partial \theta_0^j} \quad (\text{substituting } L_{meta} = L_t = \sum_{i=1}^t \ell_i)$$

$$= \frac{\partial \left( \sum_{i=1}^t \ell_i(\theta_0^j) - \alpha \frac{\partial L_{meta}(\theta_0^j)}{\partial \theta_0^j} \frac{\partial \ell_t(\theta_0^j)}{\partial \theta_0^j} \right)}{\partial \theta_0^j}$$

$$= \frac{\partial \left( \sum_{i=1}^t \ell_i(\theta_0^j) - \alpha \frac{\partial \sum_{i=1}^t \ell_i(\theta_0^j)}{\partial \theta_0^j} \frac{\partial \ell_t(\theta_0^j)}{\partial \theta_0^j} \right)}{\partial \theta_0^j} \quad (\text{expanding } L_{meta})$$

$$= \frac{\partial \left( \sum_{i=1}^t \ell_i(\theta_0^j) - \alpha \sum_{i=1}^t \frac{\partial \ell_i(\theta_0^j)}{\partial \theta_0^j} \frac{\partial \ell_t(\theta_0^j)}{\partial \theta_0^j} \right)}{\partial \theta_0^j}$$

which is the same as the gradient of Eq. 11.

where:

$$\bar{g}_k = \frac{\partial L_{meta} \left( \theta_0^j \right)}{\partial \theta_0^j} \qquad (\text{gradient of the } \textit{meta-loss} \text{ evaluated at the initial point })$$

$$\bar{g}_{k'} = \frac{\partial}{\partial \theta_0^j} L_{inner}(\theta_0^j) \quad (\text{for } k' < k) \quad (\text{gradients of the } \textit{inner-updates} \text{ evaluated at the initial point})$$

$$\theta_{k'+1}^j = \theta_{k'}^j - \alpha g_{k'} \qquad (\text{sequence of parameter vectors})$$

$$\bar{H}_k = L''_{meta}\left(\theta_0^j\right) \qquad \text{(Hessian of the \textit{meta-loss} evaluated at the initial point)}$$

$$\bar{H}_{k'} = L''_{inner}\left(\theta_0^j\right) \quad \text{(for } k' < k \text{)} \quad \text{(Hessian of the \textit{inner-objective} evaluated at the initial point)}$$

And, in our case:

$$L_{meta} = L_t = \sum_{i=1}^{t} \ell_i$$

$$L_{inner} = \ell_t$$

**Bias in the objective**: We can see in Eq. 11 that the gradient alignment term introduces some bias, which means that the parameters don't exactly converge to the minimiser of the losses on all tasks. This has been acceptable in the CL regime since we don't aim to reach the minimiser of some stationary distribution anyway (as also mentioned in Section 4.3). If we did converge to the minimiser of say $t$ tasks at some time $j$, this minimiser would no longer be optimal as soon as we see the new task $\tau_{t+1}$. Therefore, in the limit of infinite tasks and time, ensuring low-interference between tasks will pay off much more as opposed to being able to converge to the exact minima, by allowing us to make shared progress on both previous and incoming tasks.

## C  C-MAML Algorithm

Algorithm 2 outlines the training procedure for the C-MAML algorithm we propose [3].

---
**Algorithm 2** C-MAML
---
**Input:** Network weights $\theta_0^0$, inner objective $\ell$, meta objective $L$, Inner learning rate $\alpha$, Outer learning rate $\beta$
$j \leftarrow 0$
$R \leftarrow \{\}$                                             ▷ Initialise replay-buffer
**for** $t := 1$ **to** $T$ **do**
    $(X^t, Y^t) \sim D_t$
    **for** $ep := 1$ **to** $num_{epochs}$ **do**
        **for** batch $b$ **in** $(X^t, Y^t)$ **do**
            $k \leftarrow sizeof(b)$
            $b_m \leftarrow Sample(R) \cup b$            ▷ batch of samples from $\tau_{1:t}$ for *meta-loss*
            **for** $k' = 0$ **to** $k - 1$ **do**
                Push $b[k']$ to R with some probability based on reservoir sampling
                $\theta_{k'+1}^j \leftarrow \theta_{k'}^j - \alpha \cdot \nabla_{\theta_{k'}^j} \ell_t(\theta_{k'}^j, b[k'])$     ▷ *inner-update* on each incoming sample
            **end for**
            $\theta_0^{j+1} \leftarrow \theta_0^j - \beta \cdot \nabla_{\theta_0^j} L_t(\theta_k^j, b_m)$     ▷ *outer-update* by differentiating *meta-loss*
            $j \leftarrow j + 1$
        **end for**
    **end for**
**end for**

---

## D  Inter-Task Alignment

We assume that at time $j$ during training, we are seeing samples from the streaming task $\tau_t$. It is intuitive to realise that incentivising the alignment of all $\tau_{1:t}$ with the current $\tau_t$ indirectly also incentivises the alignment amongst $\tau_{1:t-1}$ as well. To demonstrate this, we compute the mean dot product of the gradients amongst the old tasks $\tau_{1:t-1}$ as the new task $\tau_t$ is added, for $t$ varying from 2 to 11. We do this for C-MAML and La-MAML on CIFAR-100.

As can be seen in Figures 4a and 4b, the alignment stays positive and roughly constant even as more tasks are added.

(a) C-MAML

(b) La-MAML

**Figure 4:** Average dot product amongst gradients of $\tau_{1:t-1}$ as new tasks are added, for the C-MAML and La-MAML algorithms calculated over 5 runs. *x-axis* shows the streaming task ID, $t$ and *y-axis* shows the cosine similarity.

## E  Robustness

Learning rate is one of the most crucial hyper-parameters during training and it often has to be tuned extensively for each experiment. In this section we analyse the robustness of our proposed variants to their LR-related hyper-parameters on the CIFAR-100 dataset. Our three variants have different sets of these hyper-parameters which are specified as follows:

- **C-MAML**: Inner and outer update LR (scalar) for the weights ($\alpha$ and $\beta$)
- **Sync La-MAML**: Inner loop initialization value for the vector LRs ($\alpha_0$), scalar learning rate of LRs ($\eta$) and scalar learning rate for the weights in the outer update ($\beta$)
- **La-MAML**: Scalar initialization value for the vector LRs ($\alpha_0$) and a scalar learning rate of LRs ($\eta$)

La-MAML is considerably more robust to tuning compared to its variants, as can be seen in Figure 5c. We empirically observe that it only requires tuning of the initial value of the LR, while being relatively insensitive to the learning rate of the LR ($\eta$). We see a consistent trend where the increase in $\eta$ leads to an increase in the final accuracy of the model. This increase is very gradual, since across a wide range of LRs varying over 2 orders of magnitude (from 0.003 to 0.3), the difference in RA is only 6%. This means that even without tuning this parameter ($\eta$), La-MAML would have outperformed most baselines at their optimally tuned values.

As seen in Figure 5a, C-MAML sees considerable performance variation with the tweaking of both the inner and outer LR. We also see that the effects of the variations of the inner and outer LR follow very similar trends and their optimal values finally selected are also identical. This means that we could potentially tune them by doing just a 1D search over them together instead of varying both independently through a 2D grid search. The Sync version of La-MAML (Figure 5b), while being relatively insensitive to the scalar initial value $\alpha_0$ and the $\eta$, sees considerable performance variation as the outer learning rate for the weights: $\beta$ is varied. This variant has the most hyper-parameters and only exists for the purpose of ablation.

Fig. 6 shows the result of 2D grid-searches over sets of the above-mentioned hyper-parameters for C-MAML and La-MAML for a better overview.

## F  Timing Comparisons

In this section, we compare the wall-clock running times (*Retained Accuracy* (RA) versus *Time*) of La-MAML against other baselines on the CIFAR100 dataset in the multi-pass setting. For ER, iCarl and La-MAML we see an increasing tread in the RA vs Time plot with La-MAML having the best

**(a)** C-MAML: Modulation of $\alpha$ and $\beta$

**(b)** Sync: Modulation of $\alpha_0$, $\eta$ and $\beta$

**(c)** La-MAML: Modulation of $\alpha_0$ and $\eta$

**Figure 5:** Retained Accuracy vs Learning Rates plot for La-MAML and its variants. Figures are plotted by varying one of the learning rate hyperparameter while keeping the others fixed at their optimal value. The hyperparameter is varied between [0.001, 0.3].

RA at the expense of the increase in time. In contrast, both AGEM and GEM perform worse than La-MAML while also taking much more running time.

## G Experimental

We carry out hyperparameter tuning for all the approaches by performing a grid-search over the range [0.0001 - 0.3] for hyper-parameters related to the learning-rate. For the multi-pass setup we use 10 epochs for all the CL approaches. In the single pass setup, all compared approaches have a hyper-parameter called *glances* which indicates the number of gradient updates or meta-updates made on each incoming sample of data. In the *Single-Pass* (LLL) setup, it becomes essential to take multiple gradient steps on each sample (or see each sample for multiple *glances*), since once we move on to later samples, we can't revisit old data samples. The performance of the algorithms naturally increases with the increase in *glances* up to a certain point. To find the optimal number of *glances* to take over each sample, we search over the values [1,2,3,5,10]. Tables 5 and 6 lists the optimal hyperparameters for all the compared approaches. All setups used the SGD optimiser since it was found to preform better than Adam [13] (possibly due to reasons stated in Section 4.3 regarding the CL setup).

To avoid exploding gradients, we clip the gradient values of all approaches at a norm of 2.0. *Class divisions* across different tasks vary with the random seeds with which the experiments were conducted. Overall, we did not see much variability across different class splits, with the variation being within 0.5-2% of the mean reported result as can be seen from Table 3

For all our baselines, we use a constant batch-size of 10 samples from the streaming task. This batch is augmented with 10 samples from the replay buffer for the replay-based approaches. La-MAML and its variants split the batch from the streaming task into a sequence of smaller disjoint sets to take multiple ($k = 10$ for MNIST and $k = 5$ for CIFAR100/TinyImagenet) gradient steps in the *inner-loop*. In MER, each sample from the incoming task is augmented with a batch of 10 replay

(a) C-MAML: Modulation of $\alpha$ and $\beta$      (b) La-MAML: Modulation of $\alpha_0$ and $\eta$

**Figure 6:** Plots of Retained Accuracy (RA) across hyper-parameter variation for C-MAML and La-MAML. We show results of the grid search over the learning rate hyperparameters. RA decreases from red to blue. All the hyperparameters are varied between [0.001, 0.3], with the axes being in log-scale.

**Figure 7:** Retained Accuracy vs Running time (seconds) for La-MAML vs other baselines on the CIFAR100 dataset.

samples to form the batch used for the meta-update. We found very small performance gaps between the first and second-order versions of our proposed variants with performance differences in the range of 1-2% for RA. This is in line with the observation that deep neural networks have near-zero hessians since the ReLU non-linearity is linear almost everywhere [23].

**MNIST Benchmarks**: On the MNIST continual learning benchmarks, images of size 28x28 are flattened to create a 1x784 array. This array is passed on to a fully-connected neural network having two layers with 100 nodes each. Each layer uses ReLU non-linearity. The output layer uses a single head with 10 nodes corresponding to the 10 classes. In all our experiments, we use a modest replay buffer of size 200 for MNIST Rotations and Permutation and size 500 for Many Permutations.

**Real-world visual classification**: For CIFAR and TinyImageNet we used a CNN having 3 and 4 conv layers respectively with 160 3x3 filters. The output from the final convolution layer is flattened and is passed through 2 fully connected layers having 320 and 640 units respectively. All the layers are succeeded by ReLU nonlinearity. Finally, a multi-headed output layer is used for performing 5-way classification for every task. This architecture is used in prior meta-learning work [30].

For CIFAR and TinyImagenet, we allow a replay buffer of size 200 and 400 respectively which implies that each class in these dataset gets roughly about 1-2 samples in the buffer. For TinyImagenet, we split the validation set into *val* and *test* splits, since the labels in the actual test set are not released.

**Table 5:** Final hyperparameters for all compared approaches on the CIFAR and TinyImagenet benchmarks

| METHOD | PARAMETER | CIFAR-100 | | TINYIMAGENET | |
|---|---|---|---|---|---|
| | | SINGLE | MULTIPLE | SINGLE | MULTIPLE |
| ER | *LR* | 0.03 | 0.03 | 0.1 | 0.1 |
| | *Epochs/Glances* | 10 | 10 | 10 | 10 |
| IID | *LR* | - | 0.03 | - | 0.01 |
| | *Epochs/Glances* | - | 50 | - | 50 |
| ICARL | *LR* | 0.03 | 0.03 | 0.01 | 0.01 |
| | *Epochs/Glances* | 2 | 10 | 2 | 10 |
| GEM | *LR* | 0.03 | 0.03 | 0.03 | 0.03 |
| | *Epochs/Glances* | 2 | 10 | 2 | 10 |
| AGEM | *LR* | 0.03 | 0.03 | 0.01 | 0.01 |
| | *Epochs/Glances* | 2 | 10 | 2 | 10 |
| MER | *LR* $\alpha$ | 0.1 | - | 0.1 | - |
| | *LR* $\beta$ | 0.1 | - | 0.1 | - |
| | *LR* $\gamma$ | 1 | - | 1 | - |
| | *Epochs/Glances* | 10 | - | 10 | - |
| META-BGD | $\eta$ | 50 | 50 | 50 | - |
| | *std-init* | 0.02 | 0.02 | 0.02 | - |
| | $\beta_{inner}$ | 0.1 | 0.1 | 0.1 | - |
| | *mc-iters* | 2 | 2 | 2 | - |
| | *Epochs/Glances* | 3 | 10 | 3 | - |
| C-MAML | $\alpha$ | 0.03 | 0.03 | 0.03 | 0.03 |
| | $\beta$ | 0.03 | 0.03 | 0.03 | 0.03 |
| | *Epochs/Glances* | 5 | 10 | 2 | 10 |
| LA-ER | $\alpha_0$ | 0.1 | 0.1 | 0.03 | 0.03 |
| | $\eta$ | 0.1 | 0.1 | 0.1 | 0.1 |
| | *Epochs/Glances* | 1 | 10 | 2 | 10 |
| SYNC LA-MAML | $\alpha_0$ | 0.1 | 0.1 | 0.075 | 0.075 |
| | $\beta$ | 0.1 | 0.1 | 0.075 | 0.075 |
| | $\eta$ | 0.3 | 0.3 | 0.25 | 0.25 |
| | *Epochs/Glances* | 5 | 10 | 2 | 10 |
| LA-MAML | $\alpha_0$ | 0.1 | 0.1 | 0.1 | 0.1 |
| | $\eta$ | 0.3 | 0.3 | 0.3 | 0.3 |
| | *Epochs/Glances* | 10 | 10 | 2 | 10 |

## H Baselines

On the MNIST benchmarks, we compare our algorithm against the baselines used in [22], which are as follows:

- Online: A baseline for the LLL setup, where a single network is trained one example at a time with SGD.

- EWC [14]: Elastic Weight Consolidation is a regularisation based method which constraints the weights important for the previous tasks to avoid catastrophic forgetting.

- GEM [17]: Gradient Episodic Memory does constrained optimisation by solving a quadratic program on the gradients of new and replay samples, trying to make sure that these gradients do not alter the past tasks' knowledge.

- MER [22]: Meta Experience Replay samples i.i.d data from a replay memory to meta-learn model parameters that show increased gradient alignment between old and current samples. We evaluate against this baseline only in the LLL setups.

On the real-world visual classification dataset, we carry out experiments on GEM, MER along with:-

**Table 6:** Final hyperparameters used for our variants on the MNIST benchmarks

| METHOD | PARAMETER | PERMUTATIONS | ROTATIONS | MANY |
|---|---|---|---|---|
| C-MAML | $\alpha$ | 0.03 | 0.1 | 0.03 |
| | $\beta$ | 0.1 | 0.1 | 0.15 |
| | *Glances* | 5 | 5 | 5 |
| SYNC LA-MAML | $\alpha_0$ | 0.15 | 0.15 | 0.03 |
| | $\beta$ | 0.1 | 0.3 | 0.03 |
| | $\eta$ | 0.1 | 0.1 | 0.1 |
| | *Glances* | 5 | 5 | 10 |
| LA-MAML | $\alpha_0$ | 0.3 | 0.3 | 0.1 |
| | $\eta$ | 0.15 | 0.15 | 0.1 |
| | *Glances* | 5 | 5 | 10 |

- IID: Network gets the data from all tasks in an independent and identically distributed manner, thus bypassing the issue of catastrophic forgetting completely. Therefore, IID acts as an upper bound for the RA achievable with this network.

- ER: Experience Replay uses a small replay buffer to store old data using reservoir sampling. This stored data is then replayed again along with the new data samples.

- iCARL [21]: iCARL is originally from the family of class incremental learners, which learns to classify images in the metric space. It prevents catastrophic forgetting by using a memory of exemplar samples to perform distillation from the old network weights. Since we perform experiments in a task incremental setting, we use the modified version of iCARL (as used by GEM [17]), where distillation loss is calculated only over the logits of the particular task.

- A-GEM [6]: Averaged Gradient Episodic Memory proposed to project gradients of the new task to a direction such as to avoid interference with respect to the average gradient of the old samples in the buffer.

- Meta-BGD: Bayesian Gradient Descent [32] proposes training a bayesian neural network for CL where the learning rate for the parameters (the means) are derived from their variances. We construct this baseline by equipping C-MAML with bayesian training, where each parameter in $\theta$ is now sampled from a gaussian distribution with a certain mean and variance. The inner-loop stays same as C-MAML(constant LR), but the magnitude of the meta-update to the parameters in $\theta$ is now influenced by their associated variances. The variance updates themselves have a closed form expression which depends on $m$ monte-carlo samples of the *meta-loss*, thus implying $m$ forward passes of the inner-and-outer loops (each time with a newly sampled $\theta$) to get $m$ meta-gradients.

# I  Discussion on Prior Work

In Table 7, we provide a comparative overview of various continual learning methods to situate our work better in the context of prior work.

*Prior-focused methods* face model capacity saturation as the number of tasks increase. These methods freeze weights to defy forgetting, and so penalise changes to the weights, even if those changes could potentially improve model performance on old tasks. They are also not suitable for the LLL setup (section 5), since it requires many passes through the data for every task to learn weights that are optimal enough to be frozen. Additionally, the success of weight freezing schemes can be attributed to over-parameterisation in neural networks, leading to sub-networks with sufficient capacity to learn separate tasks. However continual-learning setups are often motivated in resource-constrained settings requiring efficiency and scalability. Therefore solutions that allow light-weight continual learners are desirable. Meta-learning algorithms are able to exploit even small models to learn a good initialization where gradients are aligned across tasks, enabling shared progress on optimisation of task-wise objectives. Our method additionally allows meta-learning to also achieve a prior-focusing affect through the async-meta-update, without necessarily needing over-parameterised models.

In terms of resources, *meta-learning based methods* require smaller replay memories than traditional methods because they learn to generalise better across and within tasks, thus being sample-efficient. Our learnable learning rates incur a memory overhead equal to the parameters of the network. This is comparable to or less than many prior-based methods that store between 1 to $T$ scalars per parameter depending on the approach ($T$ is the number of tasks).

It should be noted that our learning rate modulation involves clipping updates for parameters with non-aligning gradients. In this aspect, it is related to methods like GEM and AGEM mentioned before. Where the distinction lies, is that our method takes some of the burden off of the clipping, by ensuring that gradients are more aligned in the first place. This means that there should be less interference and therefore less clipping of updates deemed essential for learning new tasks, on the whole.

**Table 7: Setups in prior work:** We describe the setups and assumptions adopted by prior work, focusing on approaches relevant to our method. FWT and BWT refer to forward and backward transfer as defined in [17]. '-' refers to no inductive bias for or against the specific property. Saturation of capacity refers to reduced network plasticity due to weight change penalties gradually making further learning impossible. The LLL setup is defined in Section 5. $<$ and $>$ with replay indicate that a method's replay requirements are lesser or more compared to other methods in the table. *Fishers* refers to the Fisher Information Matrix (FIM) computed per task. Each FIM has storage equal to that of the model parameters. Approaches using Bayesian Neural Networks require twice as many parameters (as does La-MAML) to store the mean and variance estimates per parameter.

| APPROACH | TRANSFER | | CAPACITY | RESOURCES | | ALGORITHM |
|---|---|---|---|---|---|---|
| | FWT | BWT | SATURATES | LLL | STORAGE | |
| PRIOR-FOCUSED | - | × | √ | × | T FISHERS | EWC [14] |
| PRIOR FOCUSED | - | × | √ | × | T MASKS | HAT [26] |
| PRIOR FOCUSED | - | × | √ | √ | 2X PARAMS | BGD/UCB [32] [8] |
| REPLAY | - | - | × | √ | > REPLAY | ICARL [21] |
| REPLAY | - | - | × | √ | > REPLAY | GEM [17] |
| META + REPLAY | √ | √ | × | √ | REPLAY | MER [22] |
| META + REPLAY | √ | √ | × | √ | REPLAY | OURS |

## Footnotes

[3]Our algorithm, *Continual-MAML* is different from a concurrent work `https://arxiv.org/abs/2003.05856` which proposes an algorithm with the same name