[Reviews · NeurIPS 2020]

Review 1

Summary and Contributions: The authors propose an extension of MAML for continual learning. Inspired by the lookahead search, the proposed method meta-learns a learning rate associate with each model parameter. The proposed method is evaluated on two image classification datasets and archives noticeable performance improvements comparing with prior work. Upate: After reading the rebuttal, I still recommend acceptance. However, I still suggest the authors to include more recent meta-learning baselines and improve the figures (e.g. font sizes and visualizations).

Strengths: - The idea of incorporating the modulation of learning rates in meta-learning is reasonable and interesting. - Experimental results of both single-pass and multiple-pass scenarios are reported. And the proposed method achieves performance improvements in both setups. - Comparisons with prior work are thoroughly discussed in the main paper and the supplementary material. - The paper is clearly written and easy to follow.

Weaknesses: - It would be better to also evaluate how the model's performances change as it sees more tasks during training. - Other types of more recent meta-learning baselines such as PEARL should be also included in the experiments. - I am curious to see if this kind of methods can be applied to reinforcement learning problems and achieves reasonable performance as well, where the training can be less robust and stable. - In Table 3, the evaluation is conducted only for 3 different seeds. It would be better to have experiments of larger scales.

Correctness: Yes. No obvious problems were found.

Clarity: The paper is clearly written and easy to follow. How the proposed method is connected to lookahead search can be better explained since it is contained in the title and the algorithm name.

Relation to Prior Work: Yes. Comparisons with prior work are thoroughly discussed in the main paper and the supplementary material.

Reproducibility: Yes

Additional Feedback: It would be better to compare with more recent meta-learning baselines and compare with baselines in the reinforcement learning setup.


Review 2

Summary and Contributions: This paper proposes Look-ahead MAML (La-MAML) for online continual learning. This method uses per-parameter learning rates for meta-learning updates and can achieve better performance than previous methods such as EWC, GEM, and MER.

Strengths: The proposed method seems novel and relevant to the NeurIPS community.

Weaknesses: I think one limitation of this work is the relatively low accuracy reported in Table 1 compared to previous work. For example, 1. In [Ref1, Ref2], the authors reported an accuracy of 84-97% for EWC and 80-93% for GEM on the MNIST Permutation benchmark. The number reported in this paper is much lower than those, only at 62% for EWC and 55% for GEM. 2. For MER, according to [Ref3], the RA is 85.50% on MNIST Permutation, which is much higher than the 73.46% reported here. 3. For GEM, according to [Ref4], the accuracy on MNIST Rotations is at 86% even when we restrict to only 1 epoch per task. The number reported in Table 1 here is only at 67.38%. There seems to be systematic differences between previously reported results and the results in this paper. The authors should explain why there are such differences. [Ref1] Nguyen et al. Variational Continual Learning. 2018. [Ref2] Swaroop et al. Improving and Understanding Variational Continual Learning. 2018. [Ref3] Riemer et al. Learning to Learn without Forgetting by Maximizing Transfer and Minimizing Interference. 2019. [Ref4] Lopez-Paz et al. Gradient Episodic Memory for Continual Learning. 2017.

Correctness: Yes, but there is a discrepancy between the results in this paper with previous results, as discussed above.

Clarity: The paper is mostly well-written. However, there are some issues below. 1. The RHS of Eq (1) does not seem to be equal to the LHS. 2. In the experiment, the paper does not specify which models are used. 3. There should be more details about the Many Permutations benchmark. 4. There should be appropriate punctuations after the equations (for example, Eq 3, 6, 7, etc).

Relation to Prior Work: Yes.

Reproducibility: No

Additional Feedback: Post-rebuttal: The rebuttal has addressed my concerns regarding the discrepancy between the results in this paper with previous results. I have increased the score and hope the authors will highlight the differences in experimental settings in the final version.


Review 3

Summary and Contributions: The authors propose a novel Meta Learning-based Continual Learning approach that expands on previously existing works (OML, MER) by reorganizing the way data is handled in meta-optimization and incorporating a learnable learning rate adjustment framework. An analysis and comparison of existing meta-learning CL methods is also provided and experiments are conducted to compare the proposed method with the state of the art.

Strengths: -The presented results are very encouraging in the task-incremental setting as intended by Hsu et al (see below). -The critical review of existing Meta-Learning CL methods is useful to form a comprehensive picture of them. It is particularly useful to establish an equivalence framework, since these works are not always easy to follow (due to complex notation).

Weaknesses: I found some issues with the experiments, that I list in the following: Line 215 states that experiments refer to “task incremental settings”. This term has a specific meaning in CL literature [3,4]: it usually means “multihead”, i.e. task labels are given at inference time. I understand that this is the setting that is featured in section 5.2. Recent literature [1, 2, 5, 6] argues that this setting is trivial and that the Single-head/Class-Incremental setting (i.e. no task labels at test time) should be preferred. Providing Class-IL results could therefore be of great help to understand how LA-MAML performs in a more challenging setting. I find the running time comparison in Table 2 not entirely satisfactory. MER is arguably among the slowest methods in CL literature. While I see the point of comparing La-MAML with its strongest competitor w.r.t. Table 1, other competitors (e.g. ER, iCaRL, EWC) are much, much faster. I believe it would be more fair to show a comparison with at least one of these methods, to let the reader understand what kind of increase in time complexity corresponds to the increase in performance brought by La-MAML. The backbone network used in section 5.2 (see appendix F lines 506 and following) is an unusual choice and I do not know of any other work adopting it. There appear to be very few convolutional layers w.r.t. ResNet (Resnet-based models are commonly used for the related datasets [1 below, 6 below, 7 below, 8 below, 6 paper references, 17 pap. references, 21 p.r.]). In addition, the fully connected layers are very large. With a backbone design this unfamiliar, I believe that the experimental results are difficult to interpret. [1] R. Aljundi et al. Gradient based sample selection for online continual learning. NIPS 2019. [2] De Lange et al. A continual learning survey: Defying forgetting in classification tasks. arXiv preprint arXiv:1909.08383, 2019. [3] G. M. van de Ven and A. S. Tolias, Three continual learning scenarios. NIPS Continual Learning Workshop, 2018. [4] YC Hsu et al. Re-evaluating continual learning scenarios: A categorization and case for strong baselines. NIPS Continual learning Workshop, 2018. [5] S. Farquhar and Y. Gal. Towards robust evaluations of continual learning. arXiv preprint arXiv:1805.09733, 2018. [6] R. Aljundi et al. Online continual learning with maximal interfered retrieval. NIPS 2019 [7] Wu et al., Large Scale Incremental Learning, CVPR 2019 [8] Hou et al., Learning a Unified Classifier Incrementally via Rebalancing, CVPR 2019

Correctness: Claims and method seem correct. I find the empirical methodology overall correct, although I have listed some criticism w.r.t. to the experimental settings.

Clarity: The paper is remarkably clear, I congratulate the authors for that.

Relation to Prior Work: Yes, there are two related sections that cover both loosely-related works and specific differences with other Meta-Learning approaches.

Reproducibility: Yes

Additional Feedback: I am happy with the author rebuttal. They answered my concern about more details in the experimental setting. I hope authors commit to revise the paper accordingly. It is ok to have a different experimental setup but it must be cristal clear in the manuscript. Given the rebuttal I raised my score of a tick.


Review 4

Summary and Contributions: Authors propose a new meta-learning based method for online continue learning.

Strengths: Their base method looks incredibly simple, i.e. aligning on average instead of pairwise gradients, yet seems to improve significantly the existing methods for online learning. Analysis is intuitive but sound. Their subsequent La-MAML is also intuitive and simple, based on an investigation of correlation between LR gradients, resulting a straightforward algorithm. Although incremental, the work has certain novelty and is highly relevant to the NeurIPS community.

Weaknesses: I can find few flaws. Perhaps the use of symbols and their super- and sub-scripts could be clearer and their meanings better explained.

Correctness: All appear sound, though I have not fully examined the appendices.

Clarity: The paper is well written indeed.

Relation to Prior Work: Fairly clear.

Reproducibility: Yes

Additional Feedback:

[Author Response · NeurIPS 2020]

We thank the reviewers for their time, feedback and highly encouraging comments. **It was acknowledged that our**
**algorithm is intuitive and principled (R4), achieves significantly better results (R1, R3), is clearly presented and**
**situated (by all), and is novel and relevant to the community (by all).** We will incorporate suggested improvements
to the paper regarding punctuation, notation, algorithm box and typos. We address here the remaining concerns:

**R2: Discrepancy in baselines' numbers, lacking experimental information**:
We appreciate the reviewer bringing up this point since it is important for the
reader to understand how our comparisons are made, which we will make clearer.
Regarding the disparity, the cited works use a higher resource allocation than
ours: a large replay buffer of size 5120 [Ref4]/5000 [Ref2 (results from AGEM)]
for GEM, while our MNIST [Rotation/Permutation] experiments have a buffer
containing only 200 samples. For EWC, the network used is 20 times larger

Figure 1: Accuracy vs Timing comparison on CIFAR (time in seconds)

[Ref2], there are only 10 tasks [Ref1, Ref2] (we have 20 tasks) and the setting is
not single-pass [Ref1] (they train each task for 20 epochs). It should be noted that
the low memory regime is where the performance trends of many CL methods
are most pronounced and meaningful. Given unlimited memory/compute, all
the methods perform comparably to ER training (as also noted by MER). We will
expand the experimental description in the main paper to highlight these details,
which we have currently outlined in Appendix F and G due to space constraints.
**R2: RHS $\neq$ LHS in Eq. 1**: While we believe the equation is technically correct, we acknowledge that it might be
confusing since we have clubbed the implied data arguments into $(X^n, Y^n)$ for brevity. We will separate these for each
of the task-specific loss functions in the camera ready for correctness.
**R3: Timing comparison**: This is a good point, and we will include a plot for our *Multi-Pass* experiments on CIFAR,
showing the total running time for La-MAML, GEM, AGEM, iCARL and ER, as shown here in Fig.1.
**R3, All: Multi-headed Problem Setting**: We thank R3 for raising this point and make a correction to our experimental
description: While our real-world vision experiments are multi-headed, our MNIST experiments are in the single-headed
domain-incremental setup (since as mentioned in Section 5.1, the output space for all tasks is the same set of 10 classes
while the common transformation to the digits varies with each task). The paper thus contains **both task-aware and**
**task-agnostic** experiments since our algorithm's working is task-agnostic. We omit Class-IL settings since they have
many of their challenges arising primarily from the bias imbalance in the classification layer. Many class-IL works
specifically focus on this issue since it is tricky to isolate it to study the general forgetting problem in CL (R3 Ref. [7]).
**R3: Choice of backbone**: We have tried to use an architecture (3-4 conv layers + 2 FCs) that is commonly used
in meta-learning works for its simplicity [1] [2] [3], and have used it to run all our baselines. In the CL setting with
ever-increasing tasks, any model will eventually be under-parameterised. As long as the model performs decently in the
*i.i.d* setting and there is a gap between the *i.i.d*-trained model and other CL methods, it should be valid to use it to study
the CL problem. If the reviewer recommends, we will add a sensitivity analysis for network sizes to the Appendix.
**R3: Relevance of LLL setting**: We agree that the issue of how setup constraints are commonly chosen in CL works
is worthy of debate. However, we reiterate that the LLL setting is challenging yet realistic in many cases where it
is not feasible to store all the within-task data points, and is also studied in many prior works like AGEM , MER. It
should also be noted that we take multiple gradient steps (*glances*) over each sample in the LLL setup (described in
Appendices E,F), thus making enough updates to the parameters.
**R3, R1: Lookahead plot**: We had hoped to show through Fig.3 in the paper, how the performance varies as more tasks
are added to the problem (as asked by R1). Note that the average accuracy stays roughly the same across an increasing
number of tasks. We shall remove this figure if it is not considered informative by the reviewers.
**R1: RL experiments**: We agree and think it would be particularly interesting to test our algorithm for Model-Based
RL, where models learnt online from a temporal data stream should undergo considerable *forgetting*. However, given
the ambiguities and lack of benchmarks for properly defining a *continual* setup in RL, we are pursuing it as an extension
and it is out of scope for this work.
**R1: Lookahead search**: We added the following: *"In optimisation literature, lookahead search usually evaluates the*
*fitness of proposed parameter updates based on an auxiliary criterion evaluated after hypothetically applying them.*
*These proposals are then modified based on evaluated fitness to make an actual update. In our case, the LRs act as*
*the modifiers of the update, and their values result from the evaluation of two criterions: the losses on old and new tasks".*
52

[1]Chelsea Finn, Aravind Rajeswaran, Sham Kakade, Sergey Levine. Online Meta-Learning: Proceedings of the 36th International Conference on Machine Learning, PMLR 97:1920-1930, 2019.

[2]Clemens Rosenbaum, Tim Klinger, and Matthew Riemer. Routing networks: Adaptive selection of non-linear functions for multi-task learning. InInternational Conference on Learning Represen-tations, 2018.

[3]Yu, T., Kumar, S., Gupta, A., Levine, S., Hausman, K., and Finn, C. Gradient surgery for multi-task learning.arXiv preprint arXiv:2001.06782, 2020.


[Meta-Review · NeurIPS 2020]

The authors proposed a simple but effective modification of MAML to adapt it to the continual setting. The analysis is also sound and experiments convincing. Finally, the rebuttal clarified questions and concerns raised by the reviewers. This is a solid contribution, timely, and of interest to the community.